# Extracting accurate materials data from research papers with conversational language models and prompt engineering

## Maciej P. Polak [1] ✉ & Dane Morgan [1] ✉

There has been a growing effort to replace manual extraction of data from research papers with automated data extraction based on natural language processing, language models, and recently, large language models (LLMs). Although these methods enable efficient extraction of data from large sets of research papers, they require a significant amount of up-front effort, expertise, and coding. In this work, we propose the `ChatExtract` method that can fully automate very accurate data extraction with minimal initial effort and background, using an advanced conversational LLM. `ChatExtract` consists of a set of engineered prompts applied to a conversational LLM that both identify sentences with data, extract that data, and assure the data's correctness through a series of follow-up questions. These follow-up questions largely overcome known issues with LLMs providing factually inaccurate responses. `ChatExtract` can be applied with any conversational LLMs and yields very high quality data extraction. In tests on materials data, we find precision and recall both close to 90% from the best conversational LLMs, like GPT-4. We demonstrate that the exceptional performance is enabled by the information retention in a conversational model combined with purposeful redundancy and introducing uncertainty through follow-up prompts. These results suggest that approaches similar to `ChatExtract`, due to their simplicity, transferability, and accuracy are likely to become powerful tools for data extraction in the near future. Finally, databases for critical cooling rates of metallic glasses and yield strengths of high entropy alloys are developed using `ChatExtract`.

Automated data extraction is increasingly used to develop databases in materials science and other fields[1]. Many databases have been created using natural language processing (NLP) and language models (LMs)[2–22]. Recently, the emergence of large language models (LLMs)[23–27] has enabled significantly greater ability to extract complex data accurately[28,29]. Previous automated methods require a significant amount of effort to set up, either preparing parsing rules (i.e., pre-defining lists of rules for identifying relevant units or particular phrases that identify the property, etc.), fine-tuning or re-training a model, or

some combination of both, which specializes the method to perform a specific task. Fine-tuning is resource and time consuming and requires extensive preparation of training data, which may not be accessible to the majority of researchers. With the emergence of conversational LLMs such as `ChatGPT`, which are broadly capable and pretrained for general tasks, there are opportunities for significantly improved information extraction methods that require almost no initial effort. These opportunities are enabled by harnessing the outstanding general language abilities of conversational LLMs, including their inherent capability to perform zero-shot (i.e., without additional training)

[1]Department of Materials Science and Engineering, University of Wisconsin-Madison, Madison, WI 53706-1595, USA. ✉e-mail: mppolak@wisc.edu; ddmorgan@wisc.edu

classification, accurate word references identification, and information retention capabilities for text within a conversation. These capabilities, combined with *prompt engineering*, which is the process of designing questions and instructions (prompts) to improve the quality of results, can result in accurate data extraction without the need for fine-tuning of the model or significant knowledge about the property for which the data is to be extracted.

Prompt engineering has now become a standard practice in the field of image generation[30–32] to ensure high quality results. It has also been demonstrated that prompt engineering is an effective method in increasing the accuracy of reasoning in LLMs[33].

In this paper we demonstrate that using conversational LLMs such as `ChatGPT` in a zero-shot fashion with a well-engineered set of prompts can be a flexible, accurate, and efficient method of extraction of materials properties in the form of the triplet *Material, Value, Unit*. We were able to minimize the main shortcoming of these conversational models, specifically errors in data extraction (e.g., improperly interpreted word relations) and hallucinations (i.e., responding with data not present in the provided text), and achieve 90.8% precision and 87.7% recall on a constrained test dataset of bulk modulus, and 91.6% precision and 83.6% recall on a full practical database construction example of critical cooling rates for metallic glasses. These results were achieved by identifying relevant sentences, asking the model to extract details about the presented data, and then checking the extracted details by asking a series of follow-up questions that suggest uncertainty of the extracted information and introduce redundancy. This approach was first demonstrated in a preprint of this paper[34], and since then a group from Microsoft has described a similar idea, but for more general tasks than materials data extraction[35]. We work with short sentence clusters made up of a target sentence, the preceding sentence, and the title, as we have found these almost always contain the full *Material, Value, Unit* triplet of data we seek. We also separated cases with single and multiple data values in a sentence, thereby greatly reducing certain types of errors. In addition, by encouraging a certain structure of responses we simplified automated post-processing the text responses into a useful database. We have put these approaches together into a single method we call `ChatExtract` —a workflow for a fully automated zero-shot approach to data extraction. We provide an example `ChatExtract` implementation in a form of a python code (see "Data availability" for more details). The prompt engineering proposed here is expected to work for essentially all *Material, Value, Unit* data extraction tasks. For different types of data extraction this prompt engineering will likely need to be modified. However, we believe that the general method, which is based on simple prompts that utilized uncertainty-inducing redundant questioning applied within an information retaining conversational model, will provide an effective and efficient approach to many types of information extraction.

The `ChatExtract` method is largely independent of the conversational LLM used and is expected to improve as the LLMs improve. Therefore, the astonishing rate of LLM improvement is likely to further support the adoption of `ChatExtract` and similar approaches to data extraction. Prompt engineering has now become a standard practice in the field of image generation[30–32] to ensure high quality results. A parallel situation may soon occur for data extraction. Specifically, a workflow such as that presented here with `ChatExtract`, which includes prompt engineering utilized in a conversational set of prompts with follow-up questions, may become a method of choice to obtain high quality data extraction results from LLMs.

## Results and discussion
### Description of the data extraction workflow
Figure 1 shows a simplified illustration of the `ChatExtract` workflow. The full workflow with all of the steps is shown in Fig. 2 so here we only summarize the key ideas behind this workflow. The initial step is preparing the data and involves gathering papers, removing html/XML syntax and dividing into sentences. This task is straightforward, standard for any data extraction effort, and described in detail in other works[29].

The data extraction is done in two main stages:

(A)   Initial classification with a simple relevancy prompt, which is applied to all sentences to weed out those that do not contain data.

(B)   A series of prompts that control the data extraction from the sentences categorized in stage (A) as positive (i.e., as relevant to the materials data at hand). To achieve high performance in Stage (B) we have developed a series of engineered prompts and the key Features of the major Stage (B) are summarized here:

(1)   Split data into single- and multi-valued, since texts containing a single entry are much more likely to be extracted properly and do not require follow up prompts, while extraction from texts containing multiple values is more prone to errors and requires further scrutinizing and verification.

(2)   Include explicitly the possibility that a piece of the data may be missing from the text. This is done to discourage the model from hallucinating non-existent data to fulfill the task.

(3)   Use uncertainty-inducing redundant prompts that encourage negative answers when appropriate. This lets the model reanalyze the text instead of reinforcing previous answers.

(4)   Embed all the questions in a single conversation as well as representing the full data in each prompt. This simultaneously takes advantage of the conversational information retention of the chat tool while each time reinforcing the text to be analyzed.

(5)   Enforce a strict Yes/No format of answers to reduce uncertainty and allow for easier automation.

Stage (A) is the first prompt given to the model. This first prompt is meant to provide information whether the sentence is relevant at all for further analysis, i.e., whether it contains the data for the property in question (value and units). This classification is crucial because, even in papers that have been extracted to be relevant by an initial keyword search, the ratio of relevant to irrelevant sentences is typically about 1:100. Therefore elimination of irrelevant sentences is a priority in the first step. Then, before starting Stage (B), we expand the text on which we are operating to a passage consisting of three sentences: the paper's *title*, the sentence *preceding* the positively classified sentence from the previous prompt, and the *positive sentence* itself. This expansion is primarily useful for making sure we include text with the material's name, which is sometimes not in the sentence targeted in Stage (A) but is most of the time in the preceding sentence or title. While in some cases the text passage built this way may not contain all the information to produce a complete datapoint, for example if the materials name is mentioned earlier in the text or in a subsection where samples are described, we found this to be a relatively rare occurrence. While technically expanding the passage to ensure extraction of complete datapoints is possible, we found that operating on as short of a text passage as possible results in the most accurate extraction, and the small gain in recall from expanding the text passage was not worth the cost of loss of precision of overall extraction. That said, tuning of the text selection approach to different LLMs and/or target properties could likely achieve improvements in some cases.

The relevant texts vary in their structure and we found it necessary to use different strategies for data extraction for those sentences that contain a single value and those sentences that contain multiple values (Feature (1) above). The texts containing only a single value are much simpler since the relation between material, value, and unit does not need to be analyzed. The LLMs tend to extract such data accurately and a single well-engineered prompt for each of the fields asked only once tends to perform very well. Texts containing multiple values

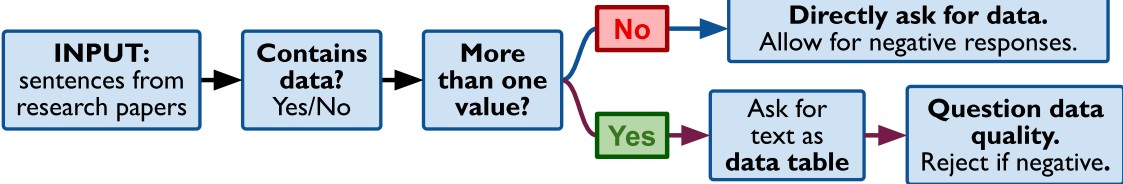

**Fig. 1 | A simplified flowchart describing our** `ChatExtract` **method of extracting structured data using a conversational large language model.** Only the key ideas for each of the steps are shown, with the fully detailed workflow presented in Fig. 2.

involve a careful analysis of the relations between words to determine which values, materials, and units correspond to one another. This complexity sometimes leads to errors in extraction or hallucinated data and requires further scrutiny and prompting with follow-up questions. Thus the first prompt in Stage (2) aims at determining whether there are multiple data points included in a given sentence, and based on that answer one of two paths is taken, different for *single-valued* and *multi-valued* sentences. As a concrete example of how often this happens, our bulk modulus dataset studied below has 70% multi-valued and 30% single-valued sentences. Our follow-up question approach proved to be very successful for the conversational `ChatGPT` models.

Next, the text is analyzed. For a single-valued text, we directly ask questions about the data in the text, asking separately for the value, its unit, and the material's name. It is important to explicitly allow for an option of a negative answer (Feature (2) above), reducing the chance that the model provides an answer even though not enough data is provided, limiting the possibility of hallucinating the data. If a negative answer is given to any of the prompt questions, the text is discarded and no data is extracted. For the case of a multi-valued sentence, instead of directly asking for data, we ask the model to provide structured data in a form of a table. This helps organize the data for further processing but can produce factually incorrect data, even if explicitly allowing negative responses. Therefore, we scrutinize each field in the provided table by asking follow-up questions (this is the redundancy of Feature (3) above) whether the data and its referencing is really included in the provided text. Again, we explicitly allow for a negative answer and, importantly, plant a seed of doubt that it is possible that the extracted table may contain some inaccuracies. Similarly as before, if any of the prompt answers are negative, we discard the sentence. It is important to notice that despite the capability of the conversational model to retain information throughout the conversation, we repetitively provide the text with each prompt (Feature (4) above). This repetition helps in maintaining all of the details about the text that is being analyzed, as the model tends to pay less attention to finer details the longer the conversation is continued. The conversational aspect and information retention improves the quality of the answers and reinforces the format of short structured answers and possibility of negative responses. The importance of the information retention in a conversation is proven later in this work by repeating our analysis exercise but with a new conversation started for each prompt, in which cases both precision and recall are significantly lowered. It is also worth noticing that we enforce a strictly *Yes* or *No* format of answers for follow up questions (Feature (5) above), which enables automation of the data extraction process. Without this constraint the model tends to answer in full sentences which are challenging to automatically analyze.

The prompts described in the flowchart (Fig. 2) are engineered by optimizing the accuracy of the responses through trial and error on various properties of varying complexity. Obviously, we have not exhausted all options, and it is likely that further optimization is possible. We have, however, noticed that contrary to intuition, providing more information about the property in the prompt usually results in worse outcomes, and we believe that the prompts

proposed here are a reliable and transferable set for many data extraction tasks.

## Performance evaluation and model comparison
We have investigated the performance of the `ChatExtract` approach on multiple property examples, including bulk modulus, metallic glass critical cooling rate, and high-entropy alloy yield stress. For bulk modulus the data is highly restricted so we can collect complete statistics on performance, and the other two cases represent applications of the method to full database generation. The bulk modulus test dataset has been chosen as a representative and particularly demanding test case for several reasons. Papers investigating mechanical properties, such as bulk modulus, very often report other elastic properties, such as the Young's modulus or shear modulus, which have similar names, ranges of values, and the same units of pressure, and are therefore easy to confuse with bulk modulus. In addition, those source documents very often describe measurements performed under pressure and other forms of stress, which have the same pressure units as bulk modulus. Finally, bulk modulus data is very often accompanied with information on the derivative of bulk modulus, which is easily confused as well. Therefore, the bulk modulus serves as a test example in which the sought property is often presented alongside numerous other, irrelevant but very easily mistakable values, presenting a challenge for accurate extraction. Our bulk modulus example data is taken from a large body of sentences extracted from hundreds of papers with bulk modulus data. To allow for an effective assessment we wanted a relatively small number of relevant sentences (containing data) of around a 100, from which we could manually extract to provide ground truth. We manually extracted data until we reached 100 relevant sentences, during which a corresponding number of 9164 irrelevant sentences (not containing data) was also labeled. We then post-processed the irrelevant sentences to remove ones that do not contain any numbers with a simple regular expression to obtain 1912 irrelevant sentences containing numbers. This preserves resources and saves time by not running the language model on sentences that obviously do not contain any data at all (since the values to be extracted are numbers), and does not impact the results of the assessment, as in our extensive tests the model never returns any datapoints from sentences that do not contain numbers. The model is explicitly instructed not to do so in the prompts (see Fig. 2), even if it mistakenly classifies the sentence as relevant in the very first classification prompt (which is very rarely the case for sentences without numbers). In these 100 sentences with data, there were a total of 179 data points (a complete triplet of material, bulk modulus, and unit combination), which we extracted by hand to serve as a ground truth dataset. We investigated the performance of multiple versions of `ChatGPT` models (see Table 1) by following the approach as described above and in Fig. 2. For true positive sentences we divide the results into categories by type of text: single-valued and multi-valued, and provide the overall performance over the entire dataset. These results are summarized in Table 1. Note that single- and multi-values columns represent performance on input passages that have data, which is of interest for understanding model behavior. The statistic that best represents the model performance on real sentences is the overall

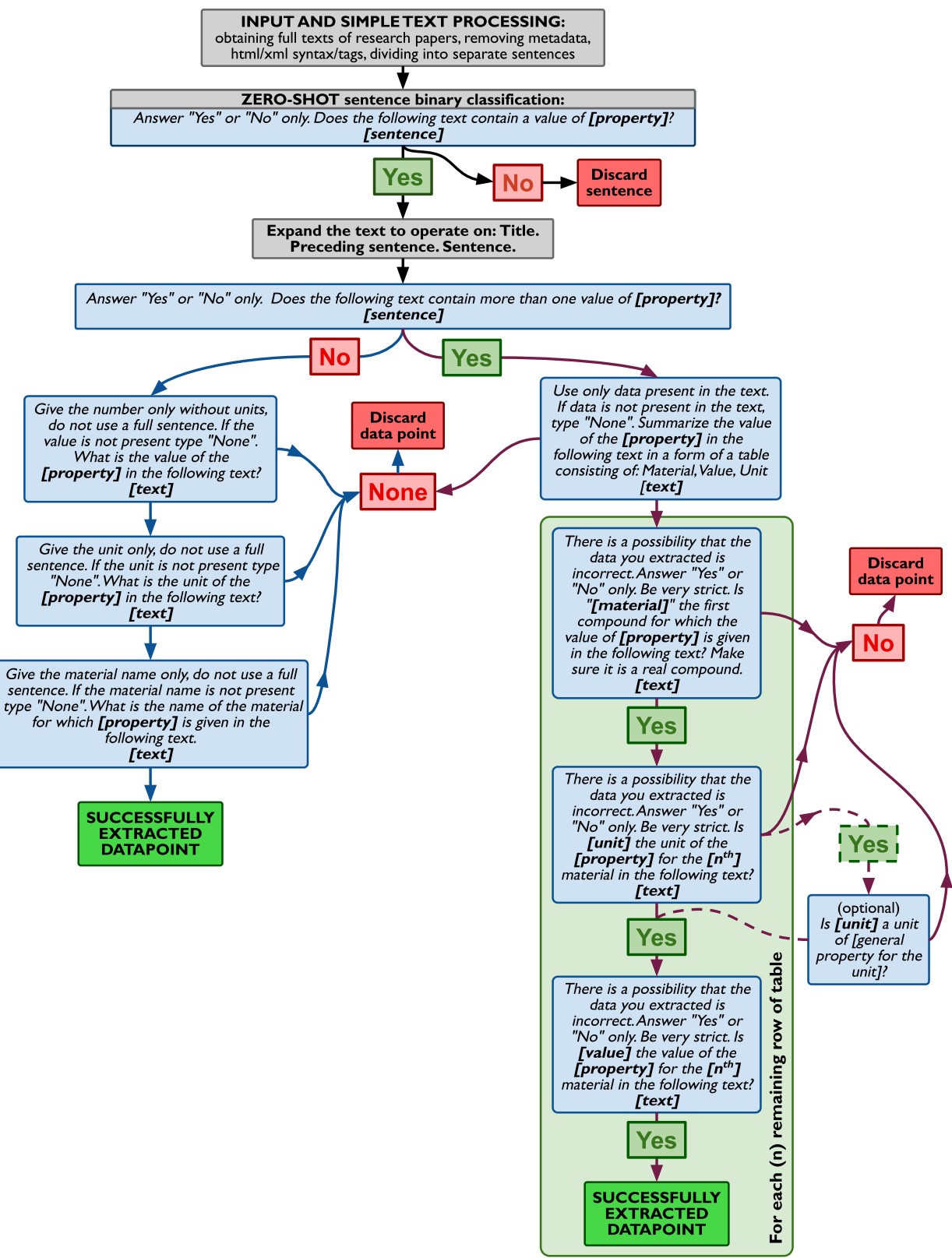

**Fig. 2 | A flowchart describing our** `ChatExtract` **method of extracting structured data using a conversational large language model.** Blue boxes represent prompts given to the model, gray boxes are instructions to the user, "Yes", "No", and "None" boxes are model's responses. The bold text in "[]" are to be replaced with appropriate values of the named item, which includes one of sentence (the target sentence being analyzed); text (the expanded text consisting of Title, Preceding sentence, and target sentence); name of the property; extracted material, value, or unit.

**Table 1 | Precision (P) and recall (R) for different types of text passages containing single- and multi-valued data, and overall, which includes all analyzed text passages, both containing data and not**

| | Single-valued | Multi-valued | Overall |
|---|---|---|---|
| **ChatGPT-4** **(gpt-4-0314)** | **P = 100%** **R = 100%** | **P = 100%** **R = 82.7%** | **P = 90.8%** **R = 87.7%** |
| **ChatGPT-3.5** **(gpt-3.5-turbo-0301)** | **P = 100%** **R = 88.5%** | **P = 97.3%** **R = 55.9%** | **P = 70.1%** **R = 65.4%** |
| **LLaMA2-chat** **(70B)** | **P = 74.1%** **R = 87.7%** | **P = 87.3%** **R = 53.5%** | **P = 61.5%** **R = 62.9%** |
| ChatGPT-4 (no follow-up) (gpt-4-0314) | P = 100% R = 100% | P = 99.2% R = 98.4% | P = 42.7% R = 98.9% |
| ChatGPT-3.5 (no follow-up) (gpt-3.5-turbo-0301) | P = 97.9% R = 88.5% | P = 94.0% R = 74.0% | P = 26.5% R = 78.2% |
| ChatGPT-3.5 (no chat) (gpt-3.5-turbo-0301) | P = 100% R = 76.9% | P = 86.6% R = 45.7% | P = 70.0% R = 54.7% |

Bold font represents final results of models used within the `ChatExtract` workflow, while the remaining demonstrate the importance of redundant follow-up questioning (no follow-up) and conversational information retention aspect (no chat).

column, where the input contains sentences both with and without data. We applied what we consider to be quite stringent criteria for assessing the performance against ground truth, the details of which an be found in the "Methods" section.

The best LLM (`ChatGPT-4`) achieved 90.8% precision at 87.7% recall, which is very impressive for a zero-shot approach that does not involve any fine-tuning. Single-valued sentences tend to be extracted with slightly higher recall (100% and 85.5% in `ChatGPT-4` and `ChatGPT-3.5`, respectively) compared to multi-valued sentences with a recall of 82.7% and 55.9% for the same models.

We believe that there are two core features of `ChatGPT` that are being used in `ChatExtract` to make this approach so successful. The first feature, and we believe the most important one, is the use of redundant prompts that introduce the possibility of uncertainty about the previously extracted data. By engineering the set of prompts and follow-up questions in this way, they substantially improve the factual correctness, and therefore precision, of the extracted information. The second feature is the conversational aspect, in which information about previous prompts and answers is retained. This allows the follow-up questions to relate to the entirety of the conversation, including the model's previous responses.

In order to demonstrate that the follow-up questions approach is essential to the good performance we repeated the exercise for both `ChatGPT` models without any follow-up questions (directly asking for structurized data only, in the same manner as before, only without asking the follow-up prompts in the multi-value branch (long light green box on right side of Fig. 2)). The results are denoted as (no follow-up) in Table 1. The dominant effects of removing follow-up questions is to allow more extracted triplets to make it to the final extracted database. This generally increases recall across all cases (single-valued, multi-valued, and overall). For passages with data (single-valued, multi-valued) these additional kept triplets are very few and almost all correct, leading to just slightly lower precisions. However, for the large number of passages with no data the additional kept triplets represent many false positives, and therefore dramatically reduce precision in the overall category. Specifically, removing follow-up questions decreases the overall precision to just 42.7% and 26.5% for `ChatGPT-4` and `ChatGPT-3.5`, respectively, from the values of 90.8% and 70.1%, respectively, resulting from a full `ChatExtract` workflow. These large reductions in precision demonstrate that follow-up questions are critical, and the analysis shows that their role is primarily to avoid the model erroneously hallucinating data in passages where none was actually given.

In order to demonstrate that the information retention provided by the conversational model is important to the good performance we repeated our approach but modified it to start a new conversation for each prompt, which meant that no conversation history was available during each prompt response. The results are denoted *(no chat)* in Table 1, This test was performed on `ChatGPT-3.5` and had little or no reduction in precision. However, there was a significant loss in recall in all categories (e.g., overall recall dropped by 10.7% to 54.7%). This loss is recall is because the multiple redundant questions tend to reject too many cases of correctly extracted triplets when the questions are asked without model knowing they are connected through a conversation. We did not perform this test on `ChatGPT-4` to reduce overall time and expense as the implications results on `ChatGPT-3.5` seemed clear.

While currently the OpenAI GPT models, in particular GPT4, are considered to be the most capable and are the most widely used, the fact that they are entirely proprietary, with a limited access dependent on OpenAI servers, and of limited transparency on their internal workings. Their default versions also tend to change their performance over time[36], which we overcome by using version snapshots (see "Methods" section), however there is no guarantee for their availability in the future. As an alternative model to assess, we chose `LLaMA2-chat` (70B)[37], a model developed by GenAI (Meta), which has extensive documentation[38], and is available to download for free and use locally without limits. The performance of the LLaMA2-chat model is summarized alongside other models in Table 1, where an overall precision and recall of 61.5 and 62.9% was achieved. The performance is close, but slightly worse than that of `ChatGPT-3.5`, which is expected based on the overall assessment of LLaMA2 capabilities[27]. While the `ChatGPT-4` model is still the most capable and performs with significantly better outcomes, this demonstrates that alternative models are also capable of data extraction, and their accuracy is likely to improve as they catch up to the state-of-the-art. It is worth noting that although the method and the prompts have been developed to be general and applicable to any LLM, the prompts have been optimized based on GPT models. Therefore, it is possible that further prompt optimization could improve performance for a specific LLM, such as LLaMA2-chat presented here, making the comparison not entirely fair for LLaMA2-chat. However, we do not expect this effect to be very significant. In order to compare the performance of ChatExtract to previous state-of-the-art data extraction methods, we performed an assessment of the performance of ChemDataExtractor2 (CDE2) on our test bulk modulus dataset. CDE2 requires, at minimum, a specifier expression and units to be explicitly specified. The specifier expression used here included all the ways we found the bulk modulus is addressed in our test data: "bulk modulus", "B", "B0", "B_0", "K", "K0", and "K_0". We also created a new unit type for units of pressure, which included all units we encountered in our test data: "GPa", "MPa", "Pa", "kbar", and "bar". CDE2 was then ran on the same text passages from our bulk modulus dataset as ChatExtract. The overall precision and recall were found to be 57% and 31% respectively, slightly lower but close to the low range results reported for thermoelectric properties (78% and 31%, respectively) obtained in ref. 6 by the authors of CDE2. We note that in this paper we use a more strict definition for a false negative datapoint than the authors of CDE2, which results in a slightly lower recall. Even though the performance of ChatExtract is better, it is worth noting that CDE2 can be efficiently executed on a personal computer with a single CPU, while the use of LLMs at the time of writing this article requires significantly higher computational power.

## Application to tables and figures

Data is not necessarily always contained within the text of the paragraph, and may be found in other structures, in particular in tables and figures. Since tables already contain structured datapoints, LLMs can certainly assist in their efficient extraction from the document. The

analysis of figures, on the other hand, is not a language processing task, and is an ongoing challenge for machine learning and artificial intelligence. LLMs can, however, help identify relevant figures for further human analysis. Figure 3 shows workflows for tables (1) and figures (2). Here, we utilize a simple workflow for table extraction—tables and their captions are gathered separately from the texts of the papers, and then they are used in classification, in a similar fashion to sentences (first step in the general ChatExtract workflow, Fig. 2 to determine whether they do contain the relevant data or not. In the case of positive classification, the text of the table and its caption are provided to the LLM and the model is instructed to only extract the relevant data for the specified property, in the form of a table, in the same way as in the general ChatExtract workflow. This step ensures that only the relevant data is extracted, as tables often contain more than just one column or one property and have to be further postprocessed. Since the data is already structured and the probability of an incorrect extraction is low, the redundant follow-up verification does not seem to be helpful and is not performed, similar to our approach for sentences for single values. For figures, only the figure caption is used in the classification, where In the case of a positive classification of a figure caption, the figure is downloaded for later manual data extraction. The accuracy for table extraction using the model which performed best for text extraction (GPT4) is quite high, as extracting structured data from an already structured table poses fewer challenges than extraction from texts. Out of 163 tables contained in the same papers which served as a source for the text bulk modulus data, we manually classified 58 as containing bulk modulus data. From these tables we were able to manually extract 500 structured bulk modulus datapoints. Using ChatExtract we were able to achieve a precision and recall for table classification of 95% and 98%, respectively. The precision and recall when counting structured data extraction for individual datapoints reached 91% and 89%, respectively. The lowering of the statistics, besides the consequence of the sporadic improper classification, was almost entirely due to an improper formatting of tables when converted from XML to raw text. While it did not happen very often, in the cases when it did, it was impossible for humans to extract data from these wrongly formatted tables as well. Even though these are not directly the method's fault, they are still counted as false positives and false negatives in our assessment, as they will inevitably be present in the final extracted database, and this is what ultimately matters the

most. Assessment of accuracy for figure classification is more difficult, as figures usually present more complex data than the simple "material, value, unit" triplets we discuss here. Therefore the criterion for a successful classification was whether the figure contained the relevant property on any of the axes, in the legend, written somewhere in the figure, or in the caption itself. Out of 436 figures contained in the same papers which served as a source for the text bulk modulus data, we manually classified 45 as containing bulk modulus data. Using the model which performed best for text extraction (GPT4) we found a 82% recall and 80% precision for the figure relevancy classification. While these results are very encouraging, it is worth noting that this is not full data extraction from figures, which is a very challenging task overall. In the case of our test bulk modulus data, for example, the bulk modulus was often contained in the pressure or energy as a function of volume plots as one of the parameters in the fitted equation of state, simply written next to the curve, while the figure caption describing the figure only says that it contains the pressure or energy as a function of volume. While a human with knowledge in the field knows that such figures represent equations of state and bulk modulus is one of the parameters in the equation of state and may expect its value in such a plot, which even a human without expertise would not be able to do. Nevertheless, in our evaluation, we considered such figures as relevant and containing data, which negatively impacted the recall. Interestingly most of the reduction in precision came from a similar reason— the figure would be explicitly captioned as containing a fitted equation of state curve, and a model would classify such a figure positively (since bulk modulus is the key parameter in the fitting) yet the figure would not directly contain the bulk modulus data.

To further demonstrate the utility of the ChatExtract approach we used the method to extract two materials property databases, one for critical cooling rates of bulk metallic glasses and one for yield strength of high entropy alloys. Before sharing the results of these data extractions it is useful to consider in more detail different types of desirable database of a materials property that might be extracted from text.

Different types of databases can be achieved with different levels of post-processing after automated data extraction. Here we describe three types of databases that we believe cover most database use cases. At one extreme is a database that encompasses all relevant mentions of a specific property, which is useful when initiating

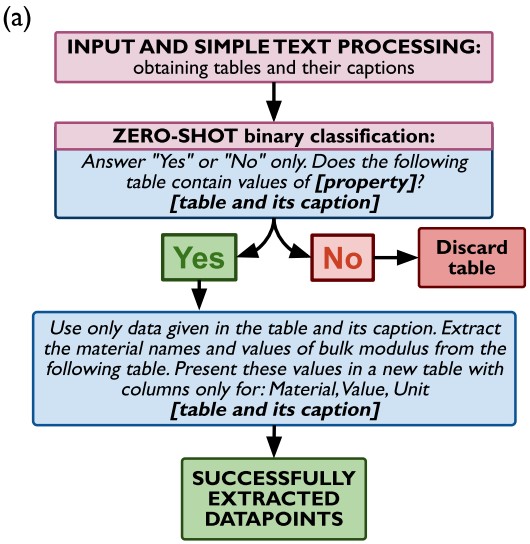

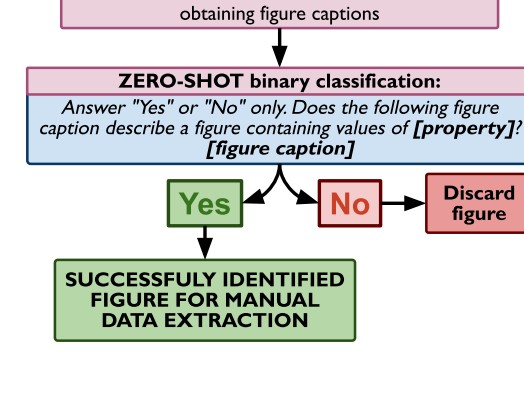

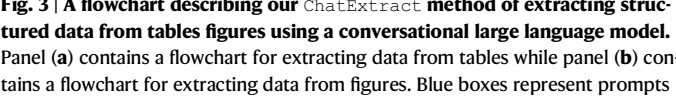

**Fig. 3 | A flowchart describing our ChatExtract method of extracting structured data from tables figures using a conversational large language model.** Panel (**a**) contains a flowchart for extracting data from tables while panel (**b**) contains a flowchart for extracting data from figures. Blue boxes represent prompts given to the model, gray boxes are instructions to the user, "Yes", "No" boxes are model's responses. The bold text in "[]" are to be replaced with appropriate values of the named item.

research in a particular field to assess the extent of data available in the literature, the span of values (including outliers), and the various material groups investigated. Entries included in such database might contain ranges, limits or approximate values, families of materials, etc. This is typically what the `ChatExtract` directly extracts, and we will refer to this as the *raw* database. At the other extreme is a strict *standardized* database, which contains uniquely defined materials with standard format, machine-readable compositions, and discrete values (i.e., not ranges or limits) with standardized units (which also helps remove the very rare occurrence where a triplet with wrong units is extracted). A *standardized* database facilitates easy interpretation and usage by computer codes and might be well-suited to, e.g., machine learning analysis. A *standardized* database can be developed from a *raw* data collection and will be a subset of that *raw* data. A third type of dataset, which is intermediate between *raw* and *standardized*, is a *cleaned* database, which removes duplicate triplets derived from within a single paper from the *raw* data, as these are almost always the exact same entry repeated multiple, e.g., in the "Discussion and Conclusions" sections) and are obviously undesirable. While the *cleaned* database can be done automatically, the *standardized* database may require some manual post-processing of the data extracted with the `ChatExtract` method.

## Results of real-life data extraction

In this study we provide two materials property databases: critical cooling rates of metallic glasses, and yield strength of high entropy alloys. Both databases are presented in all three of the above forms: *raw* data which is what is directly extracted by the `ChatExtract` approach, *cleaned* data from which single paper duplicate values have been removed, and *standardized* data where all materials that were uniquely identifiable are in a standard form of $A_X B_Y C_Z...$ (where A,B,C,... are elements and X,Y,Z,... are mole fractions). The standardization required post-processing which we accomplished manually and with a combination of further prompting with an LLM, text processing with regular expression and `pymatgen`[39]. While this standardization approach may introduce some additional errors, it provides a very useful form for the data with modest amounts of human effort. While we were able to employ further prompting combined with LLMs and text analysis tools to make this conversion, the approaches necessitate substantial additional prompt engineering and coding, which was time consuming and likely not widely applicable without significant changes. Given these limitations of our present approach to generating a *standardized* database from a *raw* database we do not discuss the details of our approach or attempt to provide a guide on how to do this most effectively. Automating the development of a *standardized* database from a *raw* database is an important topic for future work.

To limit the scope of critical cooling rates to just metallic glasses, and yield strengths to high entropy alloys, we first limited the source papers by providing a specified query to the publisher's database to return only papers relevant to the family of systems we were after, and then we applied a simple composition-based filter to the final *standardized* database. Details about both these steps are given in the following section when discussing the respective databases.

The first database is of critical cooling rates in the context of metallic glasses. In addition, the critical cooling rate database serves as a larger scale, real-life case assessment of ChatExtract, complementing the test bulk modulus dataset in evaluating the method's effectiveness. A fully manually extracted dataset of critical cooling rates was prepared to serve as ground truth to compare to the data extracted with ChatExtract. The details of the comparison and evaluation are given later in this section. The critical cooling rate dataset has been chosen due to several aspects that also make it a representative and demanding example. The critical cooling rate is often determined as a result of experimenting with different cooling rates (not *critical* cooling rates), which are values of similar magnitude and with the same units as critical cooling rates, making them very easy to confuse. In addition, critical cooling rates are sometimes described as a cooling rate for vitrification, or a cooling rate for amorphization, making extraction of the proper value very challenging. Critical cooling rate is also often present alongside other *critical* values, such as critical casting diameters. The units of critical cooling rates are also not straightforward as they often contain the degree symbol, while the unit of time in the denominator is often expressed in the form of an exponent.

To obtain source research articles we performed a search query "bulk metallic glass" + "critical cooling rate" from Elsevier's Science-Direct database which returned 684 papers, consisting of 110126 sentences.

A reference database (ground truth), which we will call $R_{c1}$, was developed using a thorough manual data extraction process based on text processing and regular expressions and aided by a previous database of critical cooling rates extracted with a more time consuming and less automated approach that involved significant human involvement[29]. This laborious process done by an experienced researcher although highly impractical, labor-intensive and time consuming, is capable of providing the most complete and accurate reference database, allowing to accurately evaluate the performance of `ChatExtract` in a real database extraction scenario, which is the most relevant assessment of the method.

To develop the critical cooling rate database with `ChatExtract`, the `ChatExtract` approach was applied identically as to the bulk modulus case except that the phrase "bulk modulus" was replaced with "critical cooling rate". We call this dataset $R_{c2}$. In comparing $R_{c2}$ data to $R_{c1}$ ground truth, the same rules for equivalency of triplet datapoints have been applied in the same way as the benchmark bulk modulus data (see "Methods"): equivalent triplets had to have identical units and values (including inequality symbols, if present), and material names had to be similar enough to allow entries to be uniquely identified as the same materials system (e.g., "$Mg_{100-x}Cu_xGd_{10}$ (x = 15)" was the same as $Mg_{85}Cu_{15}Gd_{10}$, but not the same as "Mg-Cu-Gd metallic glass" and "Zr-based bulk metallic glass" was the same as "Zr-based glass forming alloy" but not the same as $Zr_{41.2}Ti_{13.8}Cu_{12.5}Ni_{10.0}Be_{22.5}$). Critical cooling rates for bulk metallic glasses proved to be quite a challenging property to extract. The analyzed papers very often (much more often than in the other properties we worked on) contained values of critical cooling rates described as ranges or limits, and the materials were often families or broad group of materials, in particular in the "Introduction" sections of the papers. The `ChatExtract` workflow is aimed at extracting triplets of materials, value, and units without specifying further what do these mean exactly, as will be discussed in the next paragraph. To provide the most comprehensive assessment, the human curated database contains all mentions of critical cooling rates that are accompanied by any number, no matter how vague or specific. This manually extracted, very challenging *raw* database contained 721 entries. `ChatExtract` applied on the same set of research papers resulted in 634 extracted values with 76.9% precision and 63.1% recall. The vast majority of reduction in precision and recall comes from the more ambiguous material names such as the above mentioned broad groups or families of materials or ranges and limits of values. In many cases the error in extraction was minor, such as a missing inequality sign (e.g., "<0.1" in $R_{c1}$ but "0.1" in $R_{c2}$), extracting only one value from a range (e.g., "10–100" in $R_{c1}$ but only "10" in $R_{c2}$), or missing details in materials described as a group or family (e.g., "Zr-based metallic glasses" in $R_{c1}$ but only "Metallic glasses" $R_{c2}$). Even though these could be regarded as minor errors, we still consider such triplets to be incorrect. The performance is slightly improved for the *cleaned* database where a precision of 78.1% and 64.3% recall is obtained with 637 and 553 entries in $R_{c1}$ and $R_{c2}$, respectively. The most relevant *standardized* version of the database,

when extracted with `ChatExtract` yielded a final precision of 91.9% and 84.2% recall, with 313 and 286 entries subject for comparison in $R_{c1}$ and $R_{c2}$, respectively. This large reduction in the size of the *standardized* database when compared to *cleaned*, and the improvement in performance, are both due to the large amount of material groups/ families and ranges/limits of values. These cases do not classify as uniquely identifiable material compositions and discrete values so they do not satisfy the requirements for the *standardized* database, and as mentioned before, they were the most problematic for `ChatExtract` to extract (as they were for the human curating $R_{c1}$). It is important to note that in order to provide an accurate assessment of the extraction performance, as mentioned previously, the triplets are not matched by themselves, but they also have to originate from the same text passage. Therefore both the ground truth and the `ChatExtract` extracted databases were standardized separately, and if either contained a standardized value, it was considered in the assessment, making the comparison more challenging. The performance of `ChatExtract` for the *standardized* database of critical cooling rates is close to that for bulk modulus presented in Table 1 and demonstrates the transferabilty of `ChatExtract` to different properties.

In addition, we extracted 348 raw datapoints from tables, some which were duplicates of values already extracted from text data, adding only 277 new points to the standardized database and consisting of 97 new unique compositions. We also positively classified 208 figures as relevant and provided their source document and caption, but data from figures has not been manually extracted.

The final *standardized* database obtained with `ChatExtract` consists of 557 datapoints. Duplicate values originating from within a single paper have already been removed for the *cleaned* database, but duplicate triplets originating from different papers are still present. We believe it is important to keep all values, as it allows for an accurate representation of the frequency at which different systems are studied and for accurate averaging if necessary. If the duplicates were to be removed, 309 unique triplets would be left, with the many duplicates being for an industry standard system $Zr_{41.2}Ti_{13.9}Cu_{12.5}Ni_{10}Be_{22.5}$ (Vit1). The values in the final database ranged from $10^{-3}$ $Ks^{-1}$ (for $Ni_{40}P_{20}Zr_{40}$) to $4.619 \cdot 10^{13}$ $Ks^{-1}$ (for $CuZr_2$), with an average around $10^2$ $Ks^{-1}$, all quite reasonable values. An additional *standardized-MG* database is given, in which all non-metallic materials have been removed. In the case of this modest-sized database, simply removing oxygen containing systems proved to be enough and 5 non-metallic oxide materials have been removed. Out of the 309 unique datapoints, there were 222 unique material compositions (some compositions had multiple values originating from different research papers) in the *standardized* database, and after removing non-metallic systems *standardized-MG* database contained 298 unique datapoints for 217 unique material compositions. This size of 217 unique compositions is significantly larger than the previous largest hand-curated database published by Afflerbach et al.[40], which had just 77 entries. This result shows that, at least in this case, `ChatExtract` can generate more quality data with much less time than human extraction efforts. To further demonstrate the robustness of ChatExtract and compare with other methods, we applied CDE2 on the critical cooling dataset as well. CDE2 performance on the critical cooling rate dataset was consistent with the previous assessment on the bulk modulus dataset, with overall precision and recall of 49.2% and 35.1% respectively. The details on the usage of CDE2 can be found in the "Methods" section.

Finally, we developed a database of yield strength of high entropy alloys (HEAs) using the `ChatExtract` approach. This database does not have any readily available ground truth for validation but represents a very different property and alloy set than either bulk modulus or critical cooling rate and therefore further demonstrates the efficacy of the `ChatExtract` approach. In the first step we searched for a combination of the phrase "yield strength" and ("high entropy alloys" or "compositionally complex alloys" or "multi-principle component

alloys") in the Elsevier's ScienceDirect API. The search returned 4029 research papers consisting of 840431 sentences. 10269 *raw* data points were extracted. The *cleaned* database consisted of 8900 datapoints. Further post-processing yielded 4275 datapoints that constitute the *standardized* data, where we assumed that all compositions were given as atomic %, unless otherwise stated in the analyzed text (which was infrequent). The 4275 *standardized* datapoints contained a number of alloys that were not were not HEAs, with HEA defined as a systems containing 5 or more elements. The inclusion of non-HEA systems is not an error in `ChatExtract` as the data was generally in the papers, despite their being extracted by the above initial keyword search. By restricting the database to only HEAs we obtained a final *standardized-HEA* database of 2442 values. The *standardized-HEA* database had 636 materials with unique compositions. The values ranged from 12 MPa for $Al_{0.4}Co_1Cu_{0.6}Ni_1Si_{0.2}$ to 19.16 GPa for $Fe_7Cr_{31}Ni_{23}Co_{34}Mn_5$. These values are extreme but not unphysical and we have confirmed that both these extremes are extracted correctly. The distribution of yield stress values resembles a positively skewed normal distribution with a maximum around 400 MPa, which is a physically reasonable distribution shape with a peak at a typical yield stress for strong metal alloys. Additionally 2456 raw datapoints were extracted from tables, many of which were duplicates of values already extracted from text data, adding only 195 new unique HEA compositions. We positively classified 1848 figures as relevant and provide their source document and caption, but data from figures has not been manually extracted.

A large automatically extracted database of general yield strengths, not specific to HEAs, has been developed previously[5]. Direct quantitative comparison is not straightforward, but the histogram of values obtained from the previous database exhibits a very similar shape to the data obtained here, further supporting that our data is reasonable. The database of yield strengths for HEAs developed here is significantly larger than databases developed for HEAs previously, for example, databases containing yield strengths for 169 unique HEA compositions from 2018[41] and containing yield strength for 419 unique HEA compositions from 2020[42]. The `ChatExtract` generated databases are available in Figshare[43] (see "Data availability").

Now that we developed and analyzed these databases, it is easier to understand the utility of `ChatExtract`. `ChatExtract` was developed to be general and transferable, therefore it tackles a fundamental type of data extraction—a triplet of *Material, Value, Unit* for a specified property, without imposing any other restrictions. The lack of specificity when extracting "Material" or "Value" allows for extraction of data from texts where the materials are presented both as exact chemical compositions, or broad groups or families of systems. Similarly, values may be discrete numbers, or ranges or limits. However, certain restrictions are often desired in developing a database, and we believe that these fall into two broad categories with respect to the challenges of integrating them into the present `ChatExtract` workflow. The first category is restrictions based on the extracted data, for example, targeting only desired compositions or ranges of a property value. Such restrictions are trivial to integrate with `ChatExtract` by either limiting the initial search query in the publisher's database, limiting the final *standardized* database, or both, based on the restriction. For example, in our HEA database we assured only HEAs in final data by both limiting the search query in the publisher's database and applying a composition-based rule on the final *standardized* database. The second category is where we want a property value when some other property conditions hold, for example, the initial property should be considered at a certain temperature and pressure. This situation is formally straightforward for `ChatExtract` as it can be captured by generalizing the problem from finding the triplet: *material, property, unit*, to finding the multiplet: *material, property₁, unit₁, property₂, unit₂, ....* The `ChatExtract` workflow can then be generalized to apply to these multiplets by adding more steps to both the left and right branches in Fig. 2, for example if a temperature at which the data was

obtained was relevant, the left branch would contain two more boxes, the first being: *Give the number only without units, do not use a full sentence. If the value is not present type "None". What is the value of the temperature at which the value of [property] is given in the following text?*, followed by a second similar one prompting for the unit. The first prompt in the right branch would ask for a table that also included a temperature value and temperature unit, followed by two validation prompts for those two columns. This approach could be expanded into extracting non-numerical data as well, such as sample crystalinity or processing conditions. While these generalizations are formally straightforward we have made no assessment of their accuracy in this work, and some changes to `ChatExtract` might be needed to implement them effectively. For example, any additional constraints or information would have to be included in the text being examined by the LLM, and the more information that is required, the less likely it is that it will all be contained in the examined text passage. Thus the examined text passage may need to be expanded, or sometimes the required additional data may be missing from the paper altogether.

## Conclusions

This paper demonstrates that conversational LLMs such as `ChatGPT`, with proper prompt engineering and a series of follow-up questions, such as the `ChatExtract` approach presented here, are capable of providing high quality materials data extracted from research texts with no additional fine-tuning, extensive code development or deep knowledge about the property for which the data is extracted. We present such a series of well-engineered prompts and follow-up questions in this paper and demonstrate its effectiveness resulting in a best performance of over 90% precision at 87.7% recall on our test set of bulk modulus data, and 91.6% precision and 83.6% recall on a full database of critical cooling rates. We show that the success of the `ChatExtract` method lies in asking follow-up questions with purposeful redundancy and introduction of uncertainty and information retention within the conversation by comparing to results when these aspects are removed. We further develop two databases using `ChatExtract`—a database of critical cooling rates for metallic glasses and yield strengths for high entropy alloys. The first one was modest-sized and served as a benchmark for full database development since we were able to compare it to data we extracted manually. The second one was a large database, to our knowledge the largest database of yield strength of high entropy alloys to date. The high quality of the extracted data and the simplicity of the approach suggests that approaches similar to `ChatExtract` offer an opportunity to replace previous, more labor intensive, methods. Since `ChatExtract` is largely independent of the used model, it is also likely improve by simply applying it to newer and more capable LLMs as they are developed in the future.

## Methods

The main statistical quantities used to assess performance of `ChatExtract` were precision and recall, defined as:

$$\text{Precision} = \frac{\text{True Positive}}{\text{True Positive} + \text{False Positive}}$$
$$\text{Recall} = \frac{\text{True Positive}}{\text{True Positive} + \text{False Negative}}. \tag{1}$$

In our assessment we defined true positives (for precision) and false negatives (for recall) in terms of each input text passage, which we define above to consist of a target sentence, its preceding sentence, and the title. The exact approach can be confusing so we describe it concretely for every case. For a given input text passage there are zero, one or multiple unique datapoint triplets of *material, value, and unit*. We take the hand extracted triplets as the ground truth. We then process the text passage with `ChatExtract` to get a set of zero or

more extracted triplets. If the ground truth has zero triplets and the extracted data has zero triplets, this is a true negative. Every extracted triplet from a passage with zero ground truth triplets is counted as a false positive. If the ground truth has one triplet and the extracted data has zero triplets this is counted as a false negative. If the ground truth has one triplet and the extracted data has one equivalent triplet (we will define "equivalent" below) then this is counted as a single true positive. If the ground truth has one triplet and the extracted data has one inequivalent triplet then this is counted as a single false positive. If the ground truth has one triplet and the extracted data has multiple triplets they are each compared against the ground truth sequentially, assigning them as a single true positive if they are equivalent to the ground truth triplet and a single false positive if they are not equivalent to the ground truth triplet. However, only one match (a match is an equivalent pair of triplets) can be made of an extracted triplet to each ground truth triplet for a given sentence, i.e., we consider the ground truth triplet to be used up after one match. Therefore, any further extracted triplets that are equivalent to the ground truth triplet are still counted as each contributing a single false positive. Finally, we consider the case where ground truth has multiple triplets. In this case, if the extracted data has no triplets it is counted as a multiple false negatives. If the extracted data has one triplet and it is equivalent to any ground truth triplet that is counted as one true positive. If the extracted data has multiple triplets each one is compared to each ground truth triplet. If a given extracted triplet is equivalent to any one of the ground truth triplets that extracted triplet is counted as a true positive. However, as above, each ground truth triplet can only be matched once, and any addition matches of extracted triplets to an already matched ground truth triplet are counted as one additional false positive.

In the above we defined "equivalent" triplets in the following way. First, equivalent triplets had to have identical units and values (if uncertainty was present, it did not have to be extracted, but if it was extracted it had to be extracted properly as well). Second, equivalent triplets had to have materials names in the ground truth and extracted text that uniquely identified the same materials system (e.g., $Li_{17}Si_{(4-x)}Ge_x$ (x = 2.3) and $Li_{17}Si_{1.7}Ge_{2.3}$ would be equivalent but "Zr-Ni alloy" and "$Zr_{62}Ni_{38}$" would not). These requirements for equivalent triplets are quite unforgiving. In particular, in many cases where we identified false positives the LLM extracted data that was partially right or had just small errors. This fact suggests that better precision and recall might be obtained with some human input or further processing. Overall we believe the above methods provide a rigorous and demanding assessment of the `ChatExtract` approach.

OpenAI ChatGPT API was used within Python 3.10.6. To maximize reproducibility and consistency in responses we specifically used the gpt-3.5-turbo-0301 snapshot model of GPT-3.5, and gpt-4-0314 snapshot model of GPT-4, in both of which the model parameters were set as follows: temperature = 0.0, frequency_penalty = 0.0, presence_penalty = 0.0, top_p = 1.0, logprobs = False, *n* = 1, logit_bias and stop has been set to the default *null*. For the LLaMA2-chat 70B model, temperature = 0.0, and top_p = 1.0 were used, with a batch size of 6. No system prompts (empty strings) were used in any of the models. When using ChemDataExtractor2 for extracting critical cooling rates, the specifier expression was prepared based on how we found the critical cooling rate is addressed in our test data: "critical cooling rate", "Rc", "R c", "R_c", "RC", "R_C", "R C", "R c", "CCR". A new unit type for units of cooling rate was prepared, which included all units we encountered in our test data: "C/s", "K/min", "K/s", "K^(−1)", "Kmin^(−1)", "Ks-1", "Ks^(−1)", "°C/min", "°C/s", "°Cs^(−1)".

## Reporting summary

Further information on research design is available in the Nature Portfolio Reporting Summary linked to this article.

## Data availability

The extracted databases of critical cooling rates of metallic glasses and yield strength for HEAs, as well as data used in the assessment of the models is available on figshare[43]: https://doi.org/10.6084/m9.figshare.22213747. In the case of data related to figures, we do not provide the figure file due to copyrights, but instead provide the figure numbers, figure captions, and and the DOI of the source document, which allows for an easy and precise identification of the figures. In that repository, we also provide a version of the python code we used for data extraction that follows the workflow presented in Fig. 2, and involves additional simple post-processing of the `ChatGPT` responses to follow the workflow and provide a more convenient output. The post-processing included in the example code is relatively simple, and while it worked well for the properties we studied here, there may be cases when it fails in processing responses for different datasets, since occasionally `ChatGPT` may format its response in an unexpected way. The provided code is just a simple example of how `ChatExtract` could be implemented and has limited error handling.

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

## Acknowledgements
D.M. and M.P.P. acknowledge the support from National Science Foundation Cyberinfrastructure for Sustained Scientific Innovation (CSSI) Award No. 1931298. This work used Bridges-2[44] at Pittsburgh Supercomputing Center through allocation MCA09X001 from the Advanced Cyberinfrastructure Coordination Ecosystem: Services & Support (ACCESS) program, which is supported by National Science Foundation grants #2138259, #2138286, #2138307, #2137603, and #2138296.

## Author contributions
M. P. P. conceived the study, performed the modeling, tests and prepared/analyzed the results, D. M. guided and supervised the research. Writing of the manuscript was done by M. P. P. and D. M.

## Competing interests
The authors declare no competing interests.
