## [Peer Review File · Nature Communications]

Extracting Accurate Materials Data from Research Papers with Conversational Language Models and Prompt EngineeringREVIEWER COMMENTS

Reviewer #1 (Remarks to the Author):

This paper by Polak and Morgan presents an innovative approach to automated data extraction from scientific papers utilizing conversational language models like ChatGPT. The method minimizes human engineering efforts, relying on conversational prompts and follow-up questions to achieve accurate data extraction of materials properties. The method is supported by examples such as the extraction of critical cooling rates and yield strengths. I think the paper successfully demonstrates the concept of zero-shot data extraction, representing a significant advancement in materials informatics. There are a few issues I think the authors should address before it can be published:

- It would be helpful for the authors to clarify whether ChatExtract can extract information that is scattered or cross-referenced across different sentences or paragraphs. For instance, some papers may provide composition details in one section but vaguely refer to the material elsewhere.
- There is a considerable amount of data presented in tables/images, which had been challenging for traditional rule-based extraction methods due to their varying encoding formats. While the paper's focus is on textual data extraction, the authors could explore potential applications of language models like ChatGPT for extracting information from tables, images, and other non-textual elements. This would enhance the paper's practical relevance and inspire further research.
- Even though language models are usually used as black boxes, it would be necessary to discuss the potential challenges faced by the open materials research community. LLMs such as ChatGPT, remain privately held with limited transparency on their internal workings. While promoting the application of such tools, I personally found it concerning to deploy such proprietary technology, especially regarding data quality, fairness, and the overall consequences of scientific research.

Overall, I appreciate the author's efforts and look forward to the publication of the paper.

Reviewer #2 (Remarks to the Author):

Contributions:

This paper proposes a workflow to extract materials data from text using conversational LLMs in a prompt engineering way. The target triplets are elicited step by step through a series of well-designed questions and instructions. The proposed ChatExtract achieves high precision and recall scores on a synthetic dataset introduced in this paper.

Strengths:

1. The proposed ChatExtract is effective in extracting target information based on LLMs. The well-engineered prompts instruct LLMs to accurately extract the required information without producing hallucinations.
2. The proposed ChatExtract achieves high precision and recall scores on a synthetic dataset introduced in this paper.

Weaknesses:

1. This paper claims that "previous automated methods require a significant amount of effort to set up, either preparing parsing rules, ..." However, the proposed prompt-engineering pipeline can also be deemed as a rule-based approach.
2. The experiment is insufficient. It is necessary to compare ChatExtract with previous information extraction methods.
3. The synthetic dataset, containing 100 relevant sentences and 1912 irrelevant sentences, is small-scale. The synthetic dataset and code implementation are suggested to be open-sourced. Otherwise, the soundness and reproducibility of this work and doubtful.

Typos:

1. Page 1: "It has also been demonstrated that prompt engineering it an effective method" -> "It has also been demonstrated that prompt engineering is an effective method"
2. Page 1: "We work with short sentence clusters made up a a target sentence" -> "We work with short sentence clusters made up a target sentence"

Presentation Improvements:

1. To help readers better grasp the points of each part, subsections, subheadings, or subheads are encouraged to use.

Manuscript NCOMMS-23-28663
Response to Reviewers

Dear Reviewers,

Thank you for your valuable feedback on our manuscript, "*Automated Data Extraction from Scientific Papers Using Conversational Language Models*." The revisions made in response to your constructive comments have significantly improved the manuscript. We have carefully addressed all concerns raised during the review process. Among many other refinements, our main updates include expanding the method's utility for extracting data from non-textual sources, such as tables and images, integrating and evaluating open-source language models, and conducting a comparative analysis with existing methods.

We are grateful for the opportunity to refine our manuscript based on your feedback and hope that the revised version meets your expectations.

Please find a point-by-point response to all remarks below.

Best regards,

Maciej P. Polak and Dane Morgan

Reviewer #1 (Remarks to the Author):

This paper by Polak and Morgan presents an innovative approach to automated data extraction from scientific papers utilizing conversational language models like ChatGPT. The method minimizes human engineering efforts, relying on conversational prompts and follow-up questions to achieve accurate data extraction of materials properties. The method is supported by examples such as the extraction of critical cooling rates and yield strengths. I think the paper successfully demonstrates the concept of zero-shot data extraction, representing a significant advancement in materials informatics. There are a few issues I think the authors should address before it can be published:

1. It would be helpful for the authors to clarify whether ChatExtract can extract information that is scattered or cross-referenced across different sentences or paragraphs. For instance, some papers may provide composition details in one section but vaguely refer to the material elsewhere.

Response: We appreciate this comment. The current implementation of the method cannot extract widely scattered information and works only when the information

appears in the text unit we have chosen, which consists of a target sentence, the preceding sentence, and the title. This is obviously a limitation but we note some important aspects regarding this. First, while we agree that there are cases where the full information is scattered across paragraphs, during our work and dataset building process, we noticed that in the vast majority of cases, the full information could be obtained from the sentence containing the value, the sentence preceding it, and the title of the paper. This fact is supported by our high precision and recall results. Second, we note that a user can easily alter the approach to keep somewhat different text groupings, although this may not solve the general problem of widely scattered information. In particular, one obvious idea to capture data that is spread out is to simply choose a large amount of text to process at each step. Unfortunately, we found that the performance was significantly reduced with longer text units, and the present approach seemed to maximize our performance. However, tuning of the text selection approach to different LLMs and/or target properties could likely achieve improvements in some cases. The manuscript has been revised accordingly to emphasize this.

Added text (Page 2, Sec. II A):

While in some cases the text passage built this way may not contain all the information to produce a complete datapoint, for example if the materials name is mentioned earlier in the text or in a subsection where samples are described, we found this to be a relatively rare occurrence. While technically expanding the passage to ensure extraction of complete datapoints is possible, we found that operating on as short of a text passage as possible results in the most accurate extraction, and the small gain in recall from expanding the text passage was not worth the cost of loss of precision of overall extraction. That said, tuning of the text selection approach to different LLMs and/or target properties could likely achieve improvements in some cases.

2. There is a considerable amount of data presented in tables/images, which had been challenging for traditional rule-based extraction methods due to their varying encoding formats. While the paper's focus is on textual data extraction, the authors could explore potential applications of language models like ChatGPT for extracting information from tables, images, and other non-textual elements. This would enhance the paper's practical relevance and inspire further research.

Response: We agree with the reviewer and appreciate the comment, as it motivated us to significantly enhance the capabilities of our approach. In order to address this issue, we have implemented and demonstrated an additional workflow to automatically extract data from tables as well as identify figures containing data. We have added an entire new section and a new workflow figure to the paper describing the process and evaluating its performance. We performed a full assessment of the accuracy of extraction from tables and of classification of figures and discussed it in detail in the added section. Due to the now added functionality, we have also extended all of the databases (the test bulk modulus database, the critical cooling rate database, and the yield strength database) to include the table and figure data.

Added content (pages 6-7, Sec II C; pages 8.9. Sec II D):

C. Application to Tables and Figures

Data is not necessarily always contained within the text of the paragraph, and may be found in other structures, in particular in tables and figures. Since tables already contain structured datapoints, LLMs can certainly assist in their efficient extraction from the document. The analysis of figures, on the other hand, is not a language processing task, and is an ongoing challenge for machine learning and artificial intelligence. LLMs can, however, help identify relevant figures for further human analysis. Figure 4 shows workflows for tables (a) and figures (b). Here, we utilize a simple workflow for table extraction - tables and their captions are gathered separately from the texts of the papers, and then they are used in classification, in a similar fashion to sentences (first

FIG. 3. A flowchart describing our ChatExtract method of extracting structured data from: (a) tables, (b) figures; using a conversational large language model. Blue boxes represent prompts given to the model, grey boxes are instructions to the user, "Yes", "No" boxes are model's responses. The text in "[]" are to be replaced with appropriate values of the named item.

step in the general ChatExtract workflow, Fig. 3) for whether they do contain the relevant data or not. In a case of positive classification, the text of the table and its caption is provided to the LLM and the model is instructed to only extract the relevant data for the specified property, in the form of a table, in the same way as in the general ChatExtract workflow. This step ensures that only the relevant data is extracted, as tables often contain more than just one column or one property and have to be further postprocessed. Since the data is already structured and the probability of an incorrect extraction is low, the redundant follow-up verification does not seem to be helpful and is not performed, similar to our approach for sentences for single values. For figures, only the figure caption is used in the classification, where In the case of a positive classification of a figure caption, the figure is downloaded for later manual data extraction.

The accuracy for table extraction using the model which performed best for text extraction (GPT4) is quite high, as extracting structured data from an already structured table poses fewer challenges than extraction from texts. Out of 163 tables contained in

the same papers which served as a source for the text bulk modulus data, we manually classified 58 as containing bulk modulus data. From these tables we were able to manually extract 500 structured bulk modulus datapoints. Using ChatExtract we were able to achieve a precision and recall for table classification of 95% and 98%. The precision and recall when counting structured data extraction for individual datapoints reached 91% and 89%, respectively. The lowering of the statistics, besides the consequence of the sporadic improper classification, was almost entirely due to an improper formatting of tables when converted from xml to raw text. While it did not happen very often, in the cases when it did, it was impossible for humans to extract data from these wrongly formatted tables as well. Even though these are not directly the method's fault, they are still counted as false positives and false negatives in our assessment, as they will inevitably be present in the final extracted database, and this is what ultimately matters the most.

Assessment of accuracy for figure classification is more difficult, as figures usually present more complex data than the simple 'material, value, unit' triplets we discuss here. Therefore the criterion for a successful classification was whether the figure contained the relevant property on any of the axes, in the legend, written somewhere in the figure, or in the caption itself. Out of 436 figures contained in the same papers which served as a source for the text bulk modulus data, we manually classified 45 as containing bulk modulus data. Using the model which performed best for text extraction (GPT4) we found a 82% recall and 80% precision for the figure relevancy classification. While these results are very encouraging, it is worth noting that this is not full data extraction from figures, which is an extremely challenging task overall. In the case of our test bulk modulus data, for example, the bulk modulus was often contained in the pressure or energy as a function of volume plots as one of the parameters in the fitted equation of state, simply written next to the curve, while the figure caption describing the figure only says that it contains the pressure or energy as a function of volume. While a human with knowledge in the field knows that such figures represent equations of state and bulk modulus is one of the parameters in the equation of state and may expect its value in such a plot, which even a human without expertise would not be able to do. Nevertheless, in our evaluation, we considered such figures as relevant and containing data, which negatively impacted the recall. Interestingly most of the reduction in precision came from a similar reason - the figure would be explicitly captioned as containing a fitted equation of state curve, and a model would classify such a figure positively (since bulk modulus is the key parameter in the fitting) yet the figure would not directly contain the bulk modulus data.

(...) In addition, we extracted 348 raw datapoints from tables, some which were duplicates of values already extracted from text data, adding only 277 new points to the standardized database and consisting of 97 new unique compositions.

We also positively classified 208 figures as relevant and provided their source document and caption, but data from figures has not been manually extracted.

(...) Additionally 2456 raw datapoints were extracted from tables, many of which were duplicates of values already extracted from text data, adding only 195 new unique HEA compositions.

We positively classified 1848 figures as relevant and provide their source document and caption, but data from figures has not been manually extracted.

3. Even though language models are usually used as black boxes, it would be necessary to discuss the potential challenges faced by the open materials research community. LLMs such as ChatGPT, remain privately held with limited transparency on their internal workings. While promoting the application of such tools, I personally found it concerning to deploy such proprietary technology, especially regarding data quality, fairness, and the overall consequences of scientific research.

Response: We appreciate the reviewer highlighting this very valid concern. Although we have highlighted that our model was usable with any modern LLM, we had not made a serious effort to demonstrate it on the most recent open source LLMs. Therefore, in order to provide more transparency and accessibility we reevaluated our ChatExtract workflow using LLaMA2-chat (70B), a model provided by meta (facebook) that is free to use, kept constant, and more transparent. It is also considered to be one of the best free LLMs currently available. We found strong results, although somewhat less good than for GPT, which aligns with overall assessments of general capabilities of LLaMA2 as compared to GPT.

Added text (Pages 5-6, Sec II B):

While currently the OpenAI GPT models, in particular GPT4, are considered to be the most capable and are the most widely used, the fact that they are entirely proprietary, with a limited access dependent on OpenAI servers, and of limited transparency on their internal workings. Their default versions also tend to change their performance over time \cite{gpt_time}, which we overcome by using version snapshots (see Sec. IV. Methods), however there is no guarantee for their availability in the future. As an alternative model to assess, we chose LLaMA2-chat (70B) [39], a model developed by GenAI (Meta), which has extensive documentation [40], and is available to download for free and use locally without limits. The performance of the LLaMA2-chat model is summarized alongside other models in Tab I, where an overall precision and recall of 61.5% and 62.9% was achieved. The performance is close, but slightly worse than that of GPT-3.5, which is expected based on the overall assessment of LLaMA2 capabilities [39]. While the GPT-4 model is still the most capable and performs with significantly better outcomes, this demonstrates that alternative models are also capable of data extraction,

and their accuracy is likely to improve as they catch up to the state-of-the-art.

	Single-valued	Multi-valued	Overall
ChatGPT-4 (gpt-4-0314)	P=100% R=100%	P=100% R=82.7%	P=90.8% R=87.7%
ChatGPT-3.5 (gpt-3.5-turbo-0301)	P=100% R=88.5%	P=97.3% R=55.9%	P=70.1% R=65.4%
LLaMA2-chat (70B)	P=74.1% R=87.7%	P=87.3% R=53.5%	P=61.5% R=62.9%
ChatGPT-4 (no follow-up) (gpt-4-0314)	P=100% R=100%	P=99.2% R=98.4%	P=42.7% R=98.9%
ChatGPT-3.5 (no follow-up) (gpt-3.5-turbo-0301)	P=97.9% R=88.5%	P=94.0% R=74.0%	P=26.5% R=78.2%
ChatGPT-3.5 (no chat) (gpt-3.5-turbo-0301)	P=100% R=76.9%	P=86.6% R=45.7%	P=70.0% R=54.7%

Reviewer #2 (Remarks to the Author):

Contributions:

This paper proposes a workflow to extract materials data from text using conversational LLMs in a prompt engineering way. The target triplets are elicited step by step through a series of well-designed questions and instructions. The proposed ChatExtract achieves high precision and recall scores on a synthetic dataset introduced in this paper.

Strengths:

1. The proposed ChatExtract is effective in extracting target information based on LLMs. The well-engineered prompts instruct LLMs to accurately extract the required information without producing hallucinations.
2. The proposed ChatExtract achieves high precision and recall scores on a synthetic dataset introduced in this paper.

Weaknesses:

1. This paper claims that "previous automated methods require a significant amount of effort to set up, either preparing parsing rules, ..." However, the proposed prompt-engineering pipeline can also be deemed as a rule-based approach.

Response: We value the reviewer's feedback. This indeed warrants clarification - what we meant by "rule based approach" was that rules are used to perform the actual text analysis, as in searching, often without the use of any machine learning models, for particular units or phrases. Such rule-based approaches require careful tuning and knowledge for each new problem. The rules we utilize do not perform text analysis or data extraction, only guide the workflow, and can be used essentially without

modification on many different problems. We have revised the text to make this distinction clearer.

Revised text (page 1, I: Introduction):

Previous automated methods require a significant amount of effort to set up, either preparing parsing rules (i.e. pre-defining lists of rules for identifying relevant units or particular phrases that identify the property, etc.), fine-tuning or re-training a model, or some combination of both, which specializes the method to perform a specific task.

2. The experiment is insufficient. It is necessary to compare ChatExtract with previous information extraction methods.

Response: We appreciate the reviewer's concern about assessing vs. other methods. We have therefore performed an evaluation of the extracted data using another state-of-the-art materials data extraction tool, ChemDataExtractor2. The performance of ChemDataExtractor2 is worse than our ChatExtract method. The results and their discussion have been added to the text.

Added text (Pages 56, Sec II B):

In order to compare the performance of ChatExtract to previous state-of-the-art data extraction methods, we performed an assessment of the performance of ChemDataExtractor2 (CDE2) on our test bulk modulus dataset. CDE2 requires, at minimum, a specifier expression and units to be explicitly specified. The specifier expression used here included all the ways we found the bulk modulus is addressed in our test data: "bulk modulus", "B", "B0", "B_0", "K", "K0", and "K_0". We also created a new unit type for units of pressure, which included all units we encountered in our test data: "GPa", "MPa", "Pa", "kbar", and "bar". CDE2 was then ran on the same text passages from our bulk modulus dataset as ChatExtract. The overall precision and recall were found to be 57% and 31% respectively, slightly lower but close to the low range results reported for thermoelectric properties (78% and 31%, respectively) obtained in Ref. [6] by the authors of CDE2. We note that in this paper we use a more strict definition for a false negative datapoint than the authors of CDE2, which results in a slightly lower recall. Even though the performance of ChatExtract is better, it is worth noting that CDE2 can be efficiently executed on a personal computer with a single CPU, while the use of LLMs at the time of writing this article requires significantly higher computational power.

3. The synthetic dataset, containing 100 relevant sentences and 1912 irrelevant sentences, is small-scale. The synthetic dataset and code implementation are suggested to be open-sourced. Otherwise, the soundness and reproducibility of this work and doubtful.

Response: Thank you for bringing this to our attention. To clarify, we have used both a synthetic small-scale bulk modulus dataset, as well as a real-life and complete, and larger critical cooling rate dataset (721 datapoints extracted from 110126 sentences).

Datasets, as well as the full code are available to the reader on figshare, which we have updated with the new data produced as a result of this revision. We have also reworked the structure of the repository to be more readable and transparent. We have modified the text to emphasize the presence of the second dataset as well. To further improve the reproducibility, we expanded the methods section to fully reference particular snapshots of the OpenAI chat models, and we now additionally included evaluation of an alternative and publicly available LLM developed by meta (facebook) - LLaMA2-chat (please see response to point 3 for Reviewer #1).

Added text (page 8, Sec II D; page 11, Sec. V):

In addition, the critical cooling rate database serves as a larger scale, real-life case assessment of ChatExtract, complementing the test bulk modulus dataset in evaluating the method's effectiveness. A fully manually extracted dataset of critical cooling rates was prepared to serve as ground truth to compare to the data extracted with ChatExtract. The details of the comparison and evaluation are given later in this section.

(...) OpenAI ChatGPT API was used within Python 3.10.6. To maximize reproducibility and consistency in responses we specifically used the gpt-3.5-turbo-0301 snapshot model of GPT-3.5, and gpt-4-0314 snapshot model of GPT-4, in both of which the model parameters were set as follows: temperature = 0.0, frequency_penalty = 0.0, presence_penalty = 0.0, top_p = 1.0. For the LLaMA2-chat model, temperature = 0.0, and top_p = 1.0 were used. No system prompts (empty strings) were used in any of the models.

(...) The extracted databases of critical cooling rates of metallic glasses and yield strength for HEAs, as well as data used in the assessment of the models is available on figshare [43]: <https://doi.org/10.6084/m9.figshare.22213747>}

In the case of data related to figures, we do not provide the figure file due to copyrights, but instead provide the figure numbers, figure captions, and and the DOI of the source document, which allows for an easy and precise identification of the figures.

Typos:

1. Page 1: "It has also been demonstrated that prompt engineering it an effective method" -> "It has also been demonstrated that prompt engineering is an effective method"
2. Page 1: "We work with short sentence clusters made up a a target sentence" -> "We work with short sentence clusters made up a target sentence"

Response: Thank you. The text has been proofread and typos have been fixed.

Presentation Improvements:

1. To help readers better grasp the points of each part, subsections, subheadings, or subheads are encouraged to use.

Response: We thank the reviewer for this suggestion. We broke up the long *Results and Discussion* section into four subsections to make it easier for the readers to follow.

Added text (subsections of Sec. II):

- a. *Description of the Data Extraction Workflow*
- b. *Performance Evaluation and Model Comparison*
- c. *Application to Tables and Figures*
- d. *Results of Real-life Data Extraction*

REVIEWER COMMENTS

Reviewer #1 (Remarks to the Author):

I personally thank the authors for addressing my concerns comprehensively. Upon reviewing the changes, I think the authors have adequately addressed the issues. I look forward to the publication of the manuscript.

Reviewer #2 (Remarks to the Author):

While the revisions to the manuscript, particularly in terms of clarity and detail, are commendable, there are still a few critical aspects that need addressing to enhance the overall quality and impact of the work:

Comparative Analysis Depth: The comparison with ChemDataExtractor2 provides valuable context, but it seems somewhat superficial. A more in-depth analysis would be beneficial, perhaps including a variety of datasets and scenarios to showcase the robustness of ChatExtract. This would provide a clearer picture of where ChatExtract excels and where it might still need improvement.

Dataset Scale and Diversity: The inclusion of both a synthetic small-scale dataset and a real-life larger dataset is a good approach. However, the diversity of these datasets is not clearly articulated. Are these datasets representative of the variety of challenges one might encounter in real-world applications? Expanding on the types of data and the complexity involved would greatly strengthen the argument for the versatility and robustness of ChatExtract.

Methodological Transparency and Reproducibility: While the efforts to improve reproducibility are noted, the report could benefit from a more detailed explanation of the methodologies used, especially concerning the settings and parameters for the various models tested, including ChatExtract. This transparency is crucial for reproducibility and for other researchers to build upon this work.

LLM Comparison and Evaluation: The addition of an evaluation of the LLaMA2-chat model is a step forward, but it raises questions about how these different models compare in specific scenarios. A more thorough comparative analysis, highlighting the strengths and weaknesses of each model in various extraction tasks, would be informative. Also, it would be good if there are results after finetuning llama-2-chat

Manuscript NCOMMS-23-28663B
Response to Reviewers

Dear Reviewers,

Thank you for providing further valuable feedback on our manuscript, "*Automated Data Extraction from Scientific Papers Using Conversational Language Models*." We have carefully addressed the concerns raised during the review process and hope that the changes we introduced will meet your expectations.

Please find a point-by-point response to all remarks below.

Best regards,

Maciej P. Polak and Dane Morgan

Reviewer #1 (Remarks to the Author):

I personally thank the authors for addressing my concerns comprehensively. Upon reviewing the changes, I think the authors have adequately addressed the issues. I look forward to the publication of the manuscript.

Response: We thank the reviewer for their constructive criticism and we are glad that our responses were to their satisfaction.

Reviewer #2 (Remarks to the Author):

While the revisions to the manuscript, particularly in terms of clarity and detail, are commendable, there are still a few critical aspects that need addressing to enhance the overall quality and impact of the work:

1. Comparative Analysis Depth: The comparison with ChemDataExtractor2 provides valuable context, but it seems somewhat superficial. A more in-depth analysis would be beneficial, perhaps including a variety of datasets and scenarios to showcase the robustness of ChatExtract. This would provide a clearer picture of where ChatExtract excels and where it might still need improvement.

Response: To provide more analysis and variety of data for assessing ChatExtract vs. ChemDataExtractor2 (CDE2) we have applied CDE2 on a second dataset, the critical cooling rates dataset. Similarly to the previously performed assessment, CDE2 required extensive knowledge of the extracted data and time spent constructing unit and specifier files. CDE2 performance on the critical cooling rate dataset was consistent with the previous assessment on the bulk modulus dataset, with overall precision and recall of

49.2% and 35.1% respectively. On this real-life dataset, ChatExtract proved to perform very well, comparably to its performance on the test bulk modulus dataset, with 91.9% precision and 84.2% recall, significantly better than CDE2, and with no requirement for knowledge about the dataset or preparing additional task specific files. While further comparisons would certainly be of interest, each comparison requires extensive human-time intensive work in manually developing a ground truth dataset, as well as running the models and assessing the exact performance. We therefore hope that this careful comparison to a second dataset, which confirmed the outstanding performance of ChatExtract vs. CDE2, will satisfy the reviewer. While we look forward to more exploration by the community with both methods to assess their detailed strengths and weaknesses, we feel that the results demonstrating the superiority of ChatExtract are now quite clear and robust in the present work.

Added text (Page 9, Sec. II D):

To further demonstrate the robustness of ChatExtract and compare with other methods, we applied CDE2 on the critical cooling dataset as well. CDE2 performance on the critical cooling rate dataset was consistent with the previous assessment on the bulk modulus dataset, with overall precision and recall of 49.2% and 35.1% respectively. The details on the usage of CDE2 can be found in the Methods section.

Added text (Page 11, Sec. IV):

When using ChemDataExtractor2 for extracting critical cooling rates, the specifier expression was prepared based on how we found the critical cooling rate is addressed in our test data: 'critical cooling rate', 'Rc', 'R c', 'R_c', 'RC', 'R_C', 'R C', 'R c', 'CCR'. A new unit type for units of cooling rate was prepared, which included all units we encountered in our test data: 'C/s', 'K/min', 'K/s', 'K⁽⁻¹⁾', 'Kmin⁽⁻¹⁾', 'Ks⁻¹', 'Ks⁽⁻¹⁾', '°C/min', '°C/s', '°Cs⁽⁻¹⁾'.

2. Dataset Scale and Diversity: The inclusion of both a synthetic small-scale dataset and a real-life larger dataset is a good approach. However, the diversity of these datasets is not clearly articulated. Are these datasets representative of the variety of challenges one might encounter in real-world applications? Expanding on the types of data and the complexity involved would greatly strengthen the argument for the versatility and robustness of ChatExtract.

Response: We take this comment to be a request to clarify the nature of the datasets we use and how they represent diverse real-world challenges. We agree with the reviewer and appreciate the comment. We revised the description of both datasets to emphasize the difference in challenges they present to further strengthen the support for versatility and robustness of ChatExtract.

Added text (Page 3, Sec. II B):

The bulk modulus test dataset has been chosen as a representative and particularly demanding test case for several reasons. Papers investigating mechanical properties, such as bulk modulus, very often report other elastic properties, such as the Young's

modulus or shear modulus, which have similar names, ranges of values, and the same units of pressure, and are therefore easy to confuse with bulk modulus. In addition, those source documents very often describe measurements performed under pressure and other forms of stress, which have the same pressure units as bulk modulus. Finally, bulk modulus data is very often accompanied with information on the derivative of bulk modulus, which is easily confused as well. Therefore, the bulk modulus serves as a test example in which the sought property is often presented alongside numerous other, irrelevant but very easily mistakable values, presenting a challenge for accurate extraction.

Added text (Page 8, Sec. II D):

The critical cooling rate dataset has been chosen due to several aspects that also make it a representative and demanding example. The critical cooling rate is often determined as a result of experimenting with different cooling rates (not critical cooling rates), which are numbers of similar magnitude and with the same units as critical cooling rates, making them very easy to confuse. In addition, critical cooling rates are sometimes described as a cooling rate for vitrification, or a cooling rate for amorphization, making extraction of the proper value very challenging. Critical cooling rate is also often present alongside other critical values, such as critical casting diameters. The units of critical cooling rates are also not straightforward as they often contain the degree symbol, while the unit of time in the denominator is often expressed in the form of an exponent.

3. **Methodological Transparency and Reproducibility:** While the efforts to improve reproducibility are noted, the report could benefit from a more detailed explanation of the methodologies used, especially concerning the settings and parameters for the various models tested, including ChatExtract. This transparency is crucial for reproducibility and for other researchers to build upon this work.

Response: We appreciate that the reviewer has some concerns about the settings and parameters in the methods. ChatExtract on its own does not have any adjustable parameters. All details and parameters used in all of the models have been added to the Methods section in the previous revision. However, to further facilitate reproducibility, we have added a few details on the models' parameters that are available, but are by default not used, and have not been used in this research, just in case these defaults change in the future.

Added text (Page 11, Sec. IV):

To maximize reproducibility and consistency in responses we specifically used the gpt-3.5-turbo-0301 snapshot model of GPT-3.5, and gpt-4-0314 snapshot model of GPT-4, in both of which the model parameters were set as follows: temperature = 0.0, frequency_penalty = 0.0, presence_penalty = 0.0, top_p = 1.0, logprobs = False, n = 1, logit_bias and stop has been set to the default null. For the LLaMA2-chat 70B model, temperature = 0.0, and top_p = 1.0 were used, with a batch size of 6. No system prompts (empty strings) were used in any of the models.

4. LLM Comparison and Evaluation: The addition of an evaluation of the LLaMA2-chat model is a step forward, but it raises questions about how these different models compare in specific scenarios. A more thorough comparative analysis, highlighting the strengths and weaknesses of each model in various extraction tasks, would be informative. Also, it would be good if there are results after finetuning llama-2-chat

Response: While we appreciate the reviewer's suggestion for further analysis, we are unsure what "various extraction tasks" or "specific scenarios" are. In response to point 2 we expanded the description of our test dataset, emphasizing the differences between them, and the specific challenges they present, and we believe that they represent different scenarios and extraction tasks. Each dataset that is prepared as a test dataset requires separate manual extraction, a process which has to be done very carefully and takes many weeks or months, which prohibits adding more test cases in a reasonable timeframe. The consistent performance on our datasets supports the belief that the performance will be similar on other properties.

As to the last suggestion: one of the reasons ChatExtract was developed was making it accessible and easy to use, with no extensive preparation, as it is a method which leverages the capabilities of the LLMs without fine-tuning. While certainly possible, fine-tuning a model is extremely resource and time consuming and requires extensive preparation of training data, all of which are not accessible to the majority of researchers. We also believe that, despite all concerns about OpenAI, GPT-4 and whatever models will follow, will, at least for now, be the state-of-the-art and the go-to for the majority of researchers, as it undoubtedly is the most capable model available to the public, so its evaluation is the most valuable. We therefore hope that the reviewer will understand if we decline to explore methods based on fine-tuning, as they are in a very different direction than the focus on zero-shot methods and in-context learning in this work. To help make this distinction absolutely clear we have added some further clarification to the paper.

Expanded text (Page 1, Sec. I):

Previous automated methods require a significant amount of effort to set up, either preparing parsing rules (i.e. pre-defining lists of rules for identifying relevant units or particular phrases that identify the property, etc.), fine-tuning or re-training a model, or some combination of both, which specializes the method to perform a specific task.

Fine-tuning can be resource and time consuming and requires extensive preparation of training data, which may not be accessible to the majority of researchers.

(...)

*These capabilities, combined with prompt engineering, which is the process of designing questions and instructions (prompts) to improve the quality of results, can result in accurate data extraction **without the need for fine-tuning of the model or significant knowledge about the property for which the data is to be extracted.***

REVIEWERS' COMMENTS

Reviewer #2 (Remarks to the Author):

Thanks for addressing the comments. The comments are adequately addressed.